# Sustained release of ancillary amounts of testosterone and alendronate from PLGA coated pericard membranes and implants to improve bone healing

Cindy J. J. M. van de Ven[1], Nicole E. C. Bakker[2], Dennis P. Link[3], Edwin J. W. Geven[1], Jan A. Gossen[1] *

**1** Osteo-Pharma BV, Oss, The Netherlands, **2** Amphia Hospital, Breda, The Netherlands, **3** BioConnection, Oss, The Netherlands

* jan.gossen@osteo-pharma.com

**Data Availability Statement:** All relevant data are within the manuscript and its Supporting Information files.

## Abstract

Testosterone and alendronate have been identified as two bone healing compounds which, when combined, synergistically stimulate bone regeneration. This study describes the development of a novel ultrasonic spray coating for sustained release of ancillary amounts of testosterone and alendronate encapsulated in PLGA 5004A as a carrier. Due to the low amounts of testosterone and alendronate used, sensitive in vitro assays were developed to determine in vitro release. The ultrasonic spray coating technology was optimized for coating titanium screws and pericardial collagen membranes, with the aim to improve osseointegration and (guided) bone regeneration, respectively, without interfering with their primary mode of action. In vitro release analysis of collagen membranes and screws showed up to 21 days sustained release of the compounds without a burst release. Subsequent preclinical studies in rat and rabbit models indicated that testosterone and alendronate coated membranes and screws significantly improved bone regeneration in vivo. Coated membranes significantly improved the formation of new bone in a critical size calvarial defect model in rats (by 160% compared to controls). Coated screws implanted in rabbit femoral condyles significantly improved bone implant contact (69% vs 54% in controls), bone mineral density (121%) and bone volume (119%) up to 1.3 mm from the implant. Based on the results obtained, we suggest that implants or membranes enabled with local sustained delivery of ancillary amounts of testosterone and alendronate can be a promising system to stimulate local bone regeneration resulting in improved osseo-integration of implants and improved healing of bone defects and fractures.

## Introduction

Treatment of orthopedic bone defects and fractures relies on the use of medical devices such as titanium or metal screws, pins or plates to achieve proper bone healing. In addition, for

**Funding:** This work was funded by Osteo-Pharma. The funder provided support in the form of salaries for authors [CvdV, EG, NB, DL, JG], and did have a role in the study design, data collection and analysis, decision to publish, or preparation of the manuscript. The specific roles of these authors are articulated in the 'author contributions' section.

**Competing interests:** JG is CEO of Osteo-Pharma, shareholder and is inventor of patent EP3164133. CvdV and EG are employees of Osteo-Pharma. NB and DL are former employees of Osteo-Pharma and now work at BioConnection and Amphia Hospital, respectively. This does not alter our adherence to PLOS ONE policies on sharing data and materials.

treatment of dental defects, resorbable collagen membranes for guided bone regeneration have been developed and successfully applied already for many years for the treatment of defects in the jaw [1] and more recently also for orbital floor fractures [2]. To further improve the efficacy of (resorbable) implants, local sustained or controlled drug delivery of molecules that stimulate bone formation has been proposed. The goal in designing local drug delivery systems is to increase the effectiveness of the drug by localization at the site of action and thereby reducing the doses required for providing uniform systemic drug delivery. Importantly, such an approach also eliminates potential serious side effects of drugs due to systemic exposure. Many different systems for controlled or sustained drug delivery have been proposed or are in development which generally make use of various types of (biodegradable) coatings [3]. Successful applications are medical devices, such as dental screws coated with bisphosphonates to improve osseo-integration, [4], bone fillers that contain antibiotics to treat local inflammations in areas surrounding a fracture or defect [5] or drug eluting stents for cardiovascular applications [6].

In the past few years several treatment options have been investigated to improve bone regeneration. Of these, bone morphogenetic protein (BMP) 2 and 7 have been well characterized and successfully clinically tested and marketed [7]. However, a major drawback of BMPs is the high dose that is needed for successful treatment due to the short half-life of the protein. As a consequence, many side effects of the treatment have been reported varying from soft tissue swelling to incidences of infertility and cancer. Local and sustained delivery of BMPs has been proposed to overcome these limitations, but to-date has not yet been clinically tested.

Other treatment options for bone regeneration are based on the use of low molecular weight drugs such as strontium ranelate, androgens or bisphosphonates. These drugs have been successfully clinically tested for the treatment of osteoporosis [8–10], indicative for their therapeutic effects on bone cells. Unlike bisphosphonates, androgens such as testosterone, have not been approved due to the risk of potential cardiovascular side-effects as a consequence of the long-term systemic exposure required for efficacy. Testosterone, an anabolic low molecular weight compound known to positively effect muscle building, is also a potent activator of osteoblast cells and hence, bone formation. To further optimize the effect of testosterone, specifically for bone regeneration, e.g. to improve the healing of bone defects or fractures, a local sustained delivery system would be required. Moreover, bone regeneration by testosterone may be further improved by supplementing a second catabolic compound such as a bisphosphonate, which has anti-resorptive activity. Bisphosphonates bind to hydroxyapatite in the bone matrix and upon bone resorption by osteoclasts inhibits osteoclast activity, binding of bisphosphonates on bone results in prolonged biological half-live, of up to several years [11]. A synergistic effect of the combination of testosterone and alendronate has been shown in porcine femoral bone biopsies, here the combination of both compounds significantly increased osteoblast activity, which was higher than the sum of stimulation with testosterone and alendronate alone [12]. Other combination therapies of anabolic and antiresorptive compounds, including BMP2, -7 and parathyroid hormone (PTH) in combination with zoledronate or alendronate, have been shown to synergistically stimulate fracture and calvarial defect healing in preclinical models, as compared to treatment with the anabolic factor or bisphosphonate alone [13–16].

The aim of this study was to explore the use of a novel ultra-spray coating method employing PLGA 5004A as a carrier containing ancillary amounts and testosterone and alendronate to improve local bone regeneration. In this study we investigated the sustained in vitro release of testosterone to optimize the coating of collagen membranes and titanium screws. The in vivo bone regeneration capacity of coated collagen membranes and titanium screws was

subsequently analyzed in a rat critical size calvarial defect and a rabbit femoral condyle implantation model, respectively.

## Materials and methods

### Spray coating membranes

Non-sterilized porcine pericardial membranes (1x1 cm, EMCM, Nijmegen, The Netherlands) were coated on the rough side using a Prism 500 ultrasonic spray coater (USI, Haverhill, MA, USA). Membranes were coated with 4 layers of acetonitrile containing 5% PLGA 5004A (PUR-ASORB PDLG 5004A, Corbion, Amsterdam, The Netherlands), testosterone (Aspen Pharmaceuticals, Durban, South Africa) and micronized alendronate (Polpharma, Starograd Gdański, Poland) resulting in 2 or 68 $\mu g/cm^2$ of testosterone and 20 $\mu g/cm^2$ alendronate. The membranes were dried overnight in a vacuum oven at room temperature. Subsequently, the membranes were coated two or four times with 5% PLGA 5004A in acetonitrile without testosterone and alendronate and dried for 3 nights in a vacuum oven at room temperature. Afterwards membranes were packed into double Tyvek pouches. Final sterilization was performed by gamma irradiation at 25 kGy (Synergy Health, Ede, The Netherlands).

### Spray coating screws

Sandblasted screws (Ø 4.6 mm, L 10 mm, BioComp Dental B.V. Vught, The Netherlands) were coated using a Prism 500 ultrasonic spray coater. Screws were placed on a motor and were turned during spray coating. First, the screws were coated with acetonitrile containing 1% PLGA 5004A, testosterone and alendronate for 20 seconds resulting in screws with 1.5, 3 and 9 μg testosterone and 2 μg alendronate. After the acetonitrile solution was completely evaporated the screws were coated for 1 to 3 times for 20 seconds with acetonitrile containing 5% PLGA 5004A without testosterone and alendronate. Screws were dried overnight under vacuum. Coated screws were sterilized with gamma irradiation at 25 kGy (Synergy Health).

### In vitro release of testosterone and alendronate

In vitro release of testosterone from the coated membranes and screws was performed in triplicate. Membranes and screws were submerged in a vial containing 750 or 500 μl of 10 mM phosphate buffer (pH 7.4) with 0.5% SDS (elution buffer), respectively. The vials were placed in a water bath at 37°C and at predefined time points, samples of 200 μl were taken and replaced by fresh elution buffer. Testosterone concentrations from the membrane and screw samples were analyzed by HPLC (Agilent 6125, Agilent Technologies, Bad-Wurttemberg, Germany) on a ZORBAX Eclipse Plus C18 Column (Agilent Technologies) and cumulative release was determined. Alendronate concentrations from the screw samples were analyzed after a derivatization of alendronate with 2,4-dinitrofluorobenzene (DNFB) at room temperature and spectrophotometric analysis at 374 nm (Lambda 25, PerkinElmer, Waltham, MA, USA) [17].

### Surgical procedures

All surgical procedures were performed at the Radboudumc Animal Research facility (Nijmegen, The Netherlands) after the Dutch central committee on animal research and the local ethical committee on animal research of the Radboud University approved the rat critical-size calvarial defect study under project license AVD905002015159 and protocol 2017–0001 and the rabbit femoral condyle implantation study under project license AVD1030020185825 and protocol 2018–0004.

**Rat critical size calvarial defect model.** The critical-size calvarial defect surgery was performed in 12 weeks old female Wistar rats (Charles River Laboratories), and all efforts were made to minimize suffering. Rats were housed under standard laboratory conditions; per pair in standard individual ventilated cages with sawdust bedding, temperature 20–22°C, 12 h light-dark cycle, relative humidity of 45–55% and *ad libitum* access to standard rodent chow and water. To minimize postoperative pain, carprofen (4 mg/kg, Zoetis) was administered pre-operatively, 15 minutes before surgery. Anesthesia was induced and maintained by inhalation with isoflurane combined with oxygen. Rats were immobilized on their abdomen on a heating mat, the skull was shaved and disinfected with a povidone iodine solution. To minimize pain, lidocaine was administered onto the periosteum before incision. A longitudinal incision was made down to the periosteum from the nasal bone to the occipital protuberance and soft tissues were sharp dissected to visualize the calvarial periosteum. Subsequently, a midline incision was made in the periosteum and the periosteum was undermined and lifted off in left and right directions on the parietal skull. To create a central full thickness bone defect in the parietal cranium, a hollow trephine drill with an outer diameter of 8.0 mm in a dental hand piece was used. The bone defect was carefully drilled under cooling with saline, without damaging of the underlying dura. Then, the created bone segment was carefully removed, without damaging the underlying sagittal sinus. Three groups (n = 8) were randomly assigned, in the control group the defect remained empty and was not covered by a membrane, while in the other 2 groups a membrane loaded with either 2 or 68 μg testosterone in combination with 20 μg alendronate per $cm^2$ were placed over the defect. Subsequently, the skin was closed using resorbable Vicryl sutures. Carprofen was given every 24 h for a minimum of 2 days postoperatively. Animals were weighed daily up to 5 days post-surgery and weekly thereafter and were monitored daily for potential signs of dehydration, pain, infection and deviant behavior. Blood sample collection (0.5 ml) from the tail vein and μCT scanning (under anesthesia) was performed before surgery (t = 0) and after 8 weeks of implantation. The rats were euthanized at 11 weeks after surgery by $CO_2$ followed by cervical dislocation. From all animals the calvaria were isolated and fixed in 10% formalin for 48 hours and stored in 70% ethanol.

**Rabbit femoral condyle implantation model.** Mature female New Zealand White rabbits (Charles River Laboratories) with a weight of approximately 3.5 kg were used and all efforts were made to minimize suffering. Rabbits were housed under standard laboratory conditions; housing in groups of 8 with freedom of movement, temperature 20–22°C, 12 h light-dark cycle, relative humidity of 45–55%, animals were fed twice daily and had *ad libitum* access to water. Anesthesia was induced by an intramuscular injection of ketamine (10 mg/kg, Alfasan) and dexmedetomidine (100 μg/kg, Orion Pharma) and was maintained by inhalation of iso-flurane combined with oxygen. After surgery, anesthesia was antagonized by antisedan (5 mg/ml, Orion Pharma). To reduce the peri-operative infection risk, the rabbits received antibiotic prophylaxis. During anesthesia, the rabbits were immobilized on their back and the surgical areas were shaved and disinfected with povidone-iodine solution. A longitudinal incision followed by a midline incision was created in the periosteum around the knee. With several burrs (with an increasing burr diameter) the defects in the femora were drilled in the medial condyle to obtain undersized cylindrical defects. Four groups (n = 8) were randomly assigned, the control group received an uncoated screw, while in the other 3 groups coated screws were implanted containing either 1.5, 3 and 9 μg testosterone and 2 μg alendronate. After insertion of the screws, the periosteum and soft tissue were closed using resorbable Vicryl sutures. Directly after surgery and until 3 days post-operative, carprofen (4 mg/kg) and antibiotics were given every 24 h. Animals were weighed daily up to 3 days post-surgery and weekly there-after and were monitored daily for potential signs of dehydration, pain, infection and deviant behavior. During the implantation period, fluorochromes were injected subcutaneously. First,

calcein green (15 mg/kg, Sigma Aldrich) was administered 2 weeks post-surgery. Subsequently, alizarin complexone (30 mg/kg, Sigma Aldrich) was injected 4 weeks post-surgery. The rabbits were euthanized after 8 weeks post-surgery by an overdose of sodium pentobarbital and femoral condyles were harvested and fixed in 10% formalin for 48 hours and stored in 70% ethanol.

## Free testosterone ELISA

Directly after collection from the rat tail vein, blood was centrifuged (1.500 RCF, 10 min, 4˚C) and serum was aspirated and stored at -20˚C. Free testosterone concentration in serum was measured with a testosterone ELISA DM52181 (IBL International Corp, Toronto, Canada) according to the manufacturers protocol.

## MicroCT analysis

Rats were scanned in vivo with µCT (Skyscan 1076, Bruker, Kontich, Belgium) with a 1 mm aluminum filter with a resolution of 9 µm, an energy of 59 kV, an intensity of 170 mA and an integration time of 2200 ms. Rabbit femoral condyles were scanned with a resolution of 9 µm and a 1 mm aluminum filter, an energy of 100 kV, an intensity of 100 mA, and an integration time of 1200ms. The scans were reconstructed with NRecon and analyzed with CTan software. In rat calvaria defects, bone volume (BV) was assessed in a standardized volume of interest (VOI) using a predefined reference point and relative increase in BV was calculated between t = 0 and t = 8. In rabbit femoral condyles, bone mineral density (BMD), bone volume fraction (BV/TV), bone surface density (BS/TV), trabecular thickness (Tb.Th), trabecular separation (Tb.Sp) and trabecular number (Tb.N) were assessed in a standardized VOI of the trabecular bone at 0.3–0.8 mm and 0.8–1.3 mm from the screw. All values are normalized to the control group, of which the mean is set at 100%.

## Histology

Fixed rat calvaria and rabbit femoral condyle samples were processed for undecalcified histological preparation. Following dehydration in graded series of ethanol (70–100%) the samples were embedded in polymethyl-methacrylate (pMMA). After polymerization, sections from rat calvaria, 50–80 µm thick, were stained with hematoxylin-eosin (HE). The most central portion of each defect, including the section in which the defect displayed the widest extension was identified and analyzed for the presence of bone formation, new blood vessels and abnormalities such as influx of inflammatory cells. From rabbit femoral condyles thin sections (∼ 10 µm) were prepared and stained with methylene blue and basic fuchsin. Images were acquired by microscopy (Carl Zeiss Micro Imaging GmbH) and bone implant contact (BIC) was determined by using ImageJ image analysis software (Fiji distribution; National Institute of Health, Bethesda, Meryland, USA). Unstained sections were analyzed by fluorescence microscopy and fluorescent signal of calcein green and alizarin complexone was determined in a region of interest (ROI) positioned at 0.3–0.8 mm and 0.8–1.3 mm from the screw surface using ImageJ software.

## Statistical analysis

Statistical analysis was performed using GraphPad Prism 5.02 software. Data are presented as mean ± standard deviation (SD). Data were analyzed using Student's t-test at *p≤0.05, or **p≤0.01 or ***p≤0.001 compared to control.

## Results and discussion

### Optimization of ultrasonic spray coating process

PLGA 5004A was selected as polymer because of the predicted degradation time of 3 to 4 weeks. This is in line with the preferred release of testosterone over a minimal period of 3 weeks without a significant burst release (<50% at day 1) to enhance bone healing at an early stage of the fracture healing process. Sustained release of alendronate is considered less important as alendronate has the intrinsic characteristic of stable binding to hydroxyapatite in bone and is therefore not included in this study [18].

Ultrasonic spray coating of either pericard membranes or screws with only a PLGA 5004A coating solution containing testosterone and alendronate, resulted in >70% burst release of testosterone within 1 day. To prevent such a burst release and to improve the release of testosterone from the PLGA 5004A coating on pericardial membranes and orthopedic implants over longer time periods, additional layers of PLGA 5004A without alendronate and testosterone were applied on top of the coating containing alendronate and testosterone (topcoat).

For the pericardial membranes, the addition of 2 or 4 extra PLGA 5004A layers on 4 layers of PLGA 5004A containing alendronate and testosterone were tested. While the addition of 2 extra layers of PLGA 5004A without testosterone and alendronate resulted in the release of 34 ± 2% at day 1 and 72 ± 10% of the testosterone within 7 days, the addition of 4 extra PLGA 5004A layers further reduced the release of testosterone up to 16 ± 5% at day 1 and 40 ± 1% within 7 days (Fig 1A).

Similar effects were observed when adding 1 or 3 extra PLGA 5004A layers to sandblasted titanium screws coated with a single PLGA 5004A layer containing testosterone and alendronate. Adding more than 3 layers to the screw caused droplet formation and was therefore not tested. Screws were sandblasted to ensure optimal attachment of the PLGA 5004A coating onto the titanium surface. Addition of only 1 extra layer of PLGA 5004A had limited effect on the release of testosterone as 50 ± 1% of all testosterone is already released within 1 day (Fig 1B). When 2 more layers are applied (3 in total) the release of testosterone is reduced significantly up to 24 ± 1% at day 1 and 65 ± 6% within 7 days (Fig 1B). Based on these results it was decided to apply 4 and 3 additional layers of PLGA 5004A to the coated pericardial membranes and implants, respectively, to obtain a sustained release of testosterone over a period of at least 4 weeks.

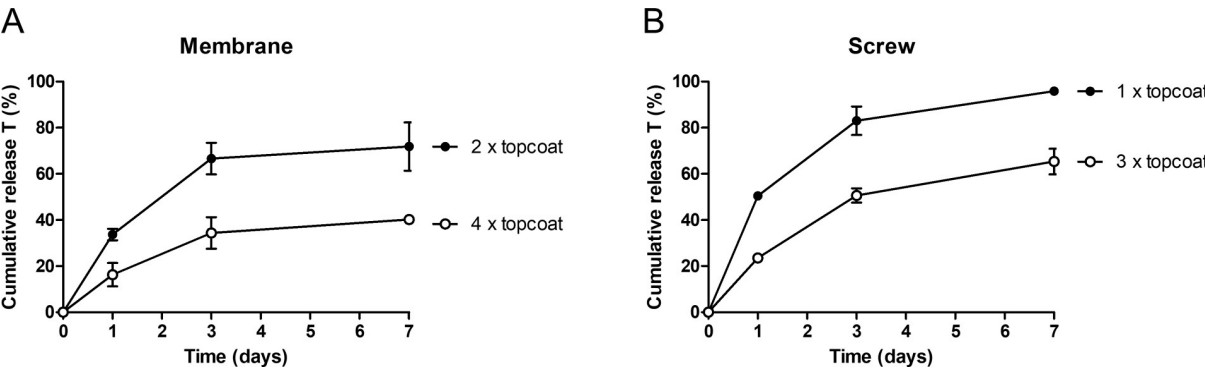

**Fig 1. Effect of additional topcoat layers of PLGA 5004A on the cumulative in vitro release of testosterone (T) from coated membranes and screws.** Addition of 2 and 4 topcoat layers on coated membranes (A, n = 2) and addition of 1 and 3 topcoat layers on coated screws (B, n = 2) reduced the in vitro release of testosterone and prevented burst release of testosterone at day 1.

### In vitro release from coated membranes and screws

Before implantation into animal models, in vitro release profiles of coated membranes and screws were determined. The in vitro release of testosterone was analyzed in membranes containing a high (68 μg/cm$^2$, Fig 2A) and a low dose (2 μg/cm$^2$, Fig 2B) of testosterone. Both membranes showed a gradual release of testosterone without an initial burst release. Release of testosterone continued over a period of 21 days, after which no additional testosterone release was detected. Alendronate release was determined in membranes containing 20 μg/cm$^2$ alendronate and a sustained release of alendronate was observed without a burst release, which continued over a period of 28 days (Fig 2C).

Similarly, screws coated with 3 different doses of testosterone (1.5, 3 and 9 μg) were analyzed for in vitro release of testosterone. For all 3 dosages a sustained release without any burst release was observed (Fig 2D). Release continued up to 21 days for each dose after which no additional release was observed. Alendronate release from screws could not be determined because of the low dose applied to the screws (2 μg) which did not reach the detection limit, however a similar release profile as observed from coated membranes (Fig 2C) is expected. Moreover, the release profile of alendronate is considered to be of lesser importance because alendronate will immediately bind to hydroxyapatite present in bone tissue and will be released upon resorption by osteoclasts. Additionally, the dose ranges of alendronate used in this study were previously determined safe and efficacious for bone regeneration in preclinical animal models [19, 20].

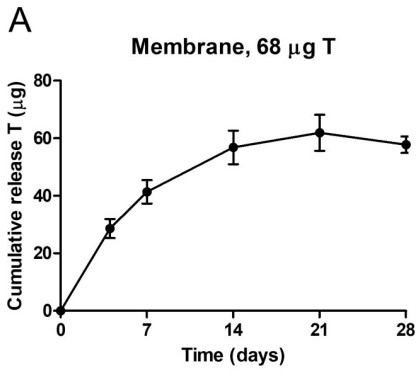

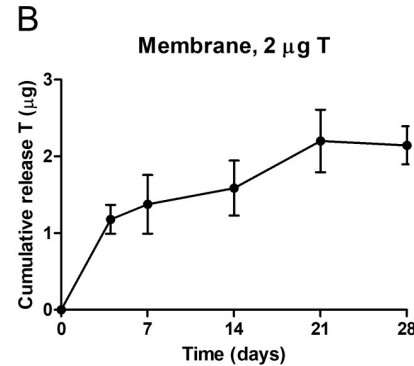

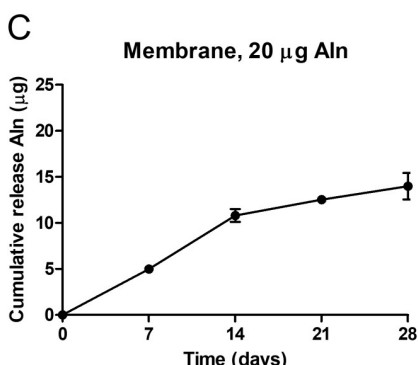

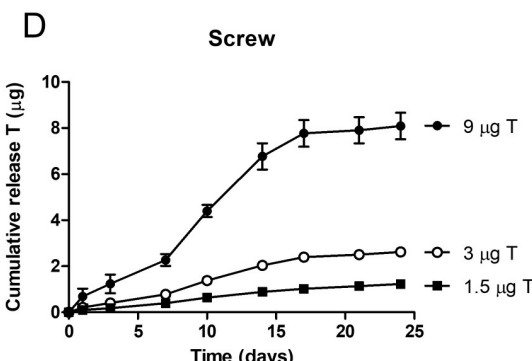

**Fig 2. Cumulative in vitro release of testosterone (T) and alendronate (Aln) from coated membranes and screws.** The release profile of testosterone from membranes coated with 68 μg (A, n = 2) and 2 μg testosterone (B, n = 2) show a sustained release of testosterone up to 21 days, while the release of alendronate continues up to 28 days (C, n = 4). The release profile of testosterone from screws coated with 1.5, 3 and 9 μg testosterone and 2 μg alendronate (D, n = 2) show a sustained release of testosterone up to 24 days.

These data indicate a sustained release of testosterone from the membrane or screw for at least 3 weeks without a significant burst release. It should be noted, however, that in vitro release using PBS + 0.5% SDS as an elution buffer may not fully represent actual in vivo release but merely will provide an accurate quality control measurement when generating different batches. Nevertheless, the release profiles obtained were considered optimal as a starting point for testing the efficacy for bone regeneration in animal models.

## Rat critical size calvarial defect

To determine the effect of the coated collagen membranes on bone healing, the coated collagen membranes were placed over an 8 mm rat critical size calvarial defect. After implantation of the membranes, µCT scanning was performed directly after the surgery (t = 0) and after 8 weeks of implantation. While membranes containing 2 µg testosterone and 20 µg alendronate did not result in significant increased bone volume compared to control, membranes containing 68 µg testosterone and 20 µg alendronate did result in a significant increase of new bone volume after 8 weeks compared to control with 160 ± 67% (p = 0.028; Fig 3A).

MicroCT analysis of the calvaria at week 8 indicated that in control animals mineralized bone growth was only observed along the edges of the defect whereas in animals treated with membranes coated with 2 and 68 µg testosterone in combination with 20 µg alendronate newly formed mineralized bone was observed covering the defect (Fig 3B). Clearly, bone formation in the control and membrane covered defects follows different bone remodelling patterns which is likely reflected by different dynamics in osteoblast and osteoclast activity. Unfortunately, the current experimental design does not allow for the differentiation of the relative effects of osteoblast activation and osteoclast inhibition as no membranes with only testosterone or alendronate were included.

Histology at week 11 confirmed this difference in bone growth patterns between control and treated defects. While control defects only showed a thin fibrous layer covering the defect (Fig 3C), newly formed bone was observed throughout the dense fibrous layer covering the defect in the treated animals (Fig 3D and 3E). Histology also revealed the formation of novel blood vessels indicating conditions that are optimal for supply of nutrients for bone growth (Fig 3F). Importantly, influx of inflammatory cells or any other histological abnormalities were not observed in any of the samples analysed indicating the absence of any adverse effects due to the treatment of the defect with a coated membrane.

Analysis of sera from animals treated with 68 µg testosterone in combination with 20 µg alendronate revealed no detectable levels of free testosterone (<0.245 pg/ml) above background levels indicating absence of systemic exposure to testosterone released from coated membranes.

Beside osteoblast activity, osteoclast activity play an important role in fracture healing, through callus resorption and bone remodelling, and therefore the long-term use of bisphosphonates has been suggested to impair fracture healing. Preclinical studies revealed that systemic bisphosphonate treatment was associated with increased callus size but no delay in fracture healing or decreased mechanical strength was observed [21]. Similarly, in humans, systemic bisphosphonate treatment did not affect fracture healing in distal radius, hip or vertebral fractures [22]. As of such we do not anticipate a negative effect of alendronate on bone formation in this study, considering the local and sustained release of ancillary amounts of alendronate. Indeed, coated membranes with similar amounts of testosterone and alendronate placed over a minipig mandibular defect resulted in increased bone formation, without disrupting bone remodelling as normal osteoblast and osteoclast activity was observed [12].

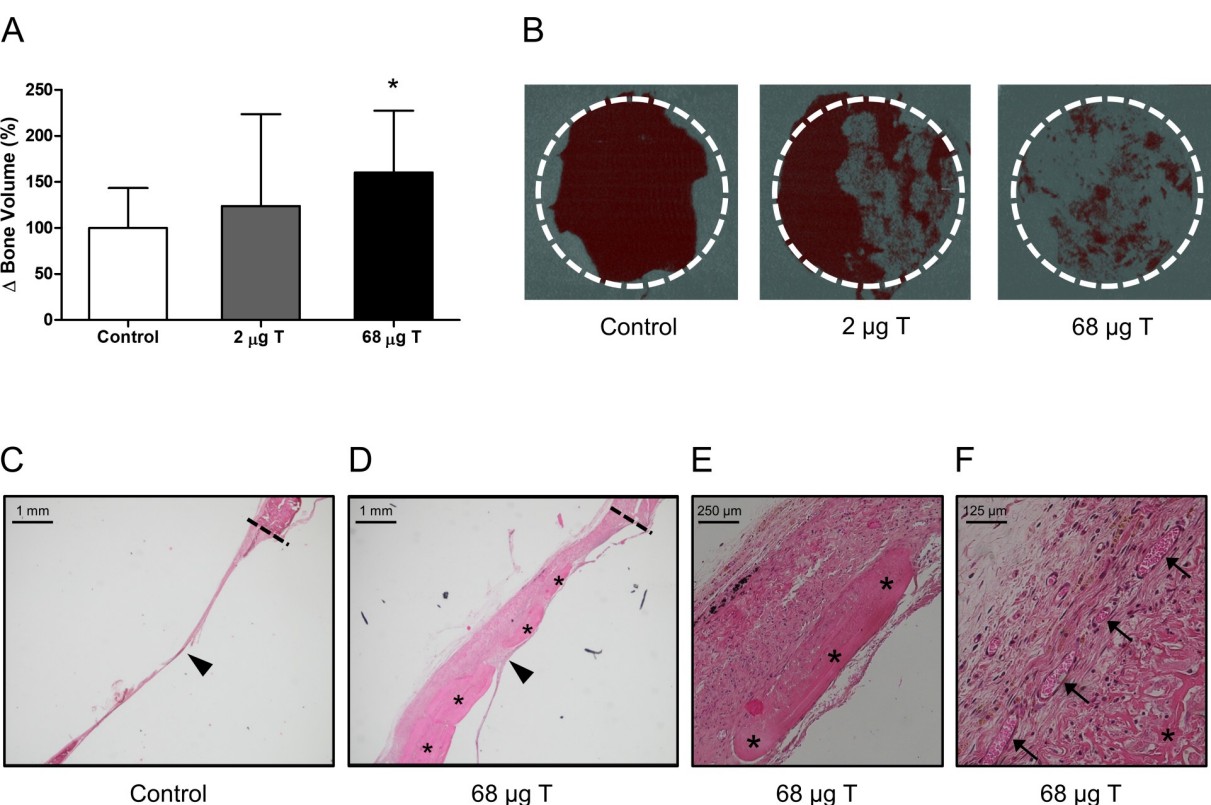

**Fig 3. Rat critical size calvarial defect model.** Defects covered with membranes coated with 68 μg testosterone (T) and 20 μg alendronate showed increased bone volume (BV) compared to controls after 8 weeks (A, n = 7–8). Two dimensional μCT images of the defects (B) indicated different bone growth patterns for controls (mainly along the defect edge (dashed line)) and membrane covered defects (covering the defect) (B). HE-staining of sections of control defects after 11 weeks showed a thin fibrous tissue layer (arrowhead) without newly formed bone connected to the defect edge (dashed line) and covering the defect (C, 2.5x). Defects covered with a membrane coated with 68 μg testosterone and 20 μg alendronate after 11 weeks showed a thick fibrous layer (arrowhead) connected to the defect edge (dashed line) in which newly formed bone is visible (asterisk) (D, 2.5x). Higher magnifications show the newly formed bone in more detail (E, 10x), and the presence of blood vessels (arrows) within the fibrous layer (F, 20x). Significant differences compared to controls are indicated by asterisks: *p≤0.05.

## Rabbit condyle model

To study the effect of the coating of screws, uncoated screws and screws coated with increasing amounts of testosterone (1.5–9 μg) and 2 μg alendronate were implanted in the femoral condyle of rabbits. After 2 and 4 weeks of implantation fluorescent dyes were injected that will be incorporated into newly formed bone. After 8 weeks, the animals were euthanized and the bone area surrounding the screw was analyzed by histology and μCT.

Analysis of the fluorescent signal was performed on histological sections. Implants were placed in the medial condyle and positioned within the subchondral trabecular bone (Fig 4A) and fluorescent microscopy of the sections showed incorporation of calcein green and alizarin complexone fluorescent dyes in the newly formed bone (Fig 4B). The fluorescent signal of both dyes was analyzed on sections of non-coated implants and implants coated with 9 μg testosterone and 2 μg alendronate. Signal was determined in ROIs positioned at 0.3–0.8 mm (ROI1) and 0.8–1.3 mm (ROI2) from the screw surface and percentage of fluorescent signal to total ROI area was calculated. No significant difference in fluorescent signal of calcein green in ROI1 (control; 4.1 ± 3.1%, 9 μg T; 4.3 ± 2.1%, p = 0.908) or ROI2 (control; 4.5 ± 2.8%, 9 μg T; 2.4 ± 2.2%, p = 0.356) and of alizarin complexone in VOI1 (control; 5.9 ± 8.9%, 9 μg T; 2.2 ± 1.5%, p = 0.517) or VOI2 (control; 3.9 ± 5.6%, 9 μg T; 0.8 ± 0.4%, p = 0.0932) was

A

B

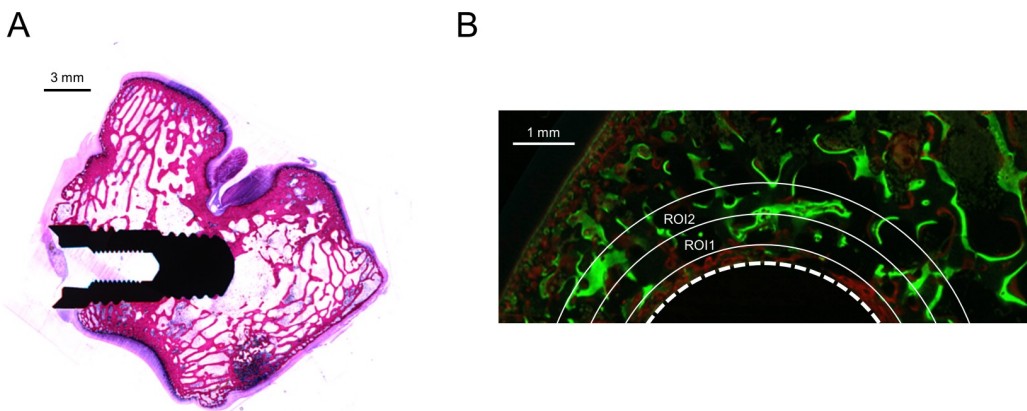

**Fig 4. Histological analysis of rabbit femoral condyle implantation model after 8 weeks.** Histological section stained with HE shows positioning of screw in medial femoral condyle within the subchondral trabecular bone (A). Incorporation of calcein green (green signal) and alizarine complexone (red signal) fluorescent dyes into newly formed bone at 2 and 4 weeks after implantation, respectively, of a screw coated with 9 µg testosterone and 2 µg alendronate (B). Fluorescent signal was observed directly at the surface of the screw (dashed line) but also extending into VOI1 (0.3–0.8 mm from screw) and VOI2 (0.8–1.3 mm from screw).

observed. The lack of a significant difference in incorporation fluorescent dyes at week 2 and week 4 after implantation may be explained by large standard deviation in fluorescent signal which may be due to small variations in the position of the screw in the subchondral bone and the position of the histological section that is analyzed. Moreover, the fluorescent signal only reflects bone formation at a single time point and may not be representative for the total bone formed at end point.

Before embedding for histology, the rabbit condyle was analyzed by µCT. The areas analyzed were 0.3–0.8 mm (VOI1) and 0.8–1.3 mm (VOI2) from the screw. Due to scattering of the screw, areas less than 0.3 mm from the screw cannot be analyzed by µCT. The data obtained (Fig 5A) for VOI1 indicated significant increases in BMD ($130 \pm 30\%$, $p = 0.012$), BS/TV ($118 \pm 15\%$, $p = 0.012$) and a significant decrease of Tb.Sp ($91 \pm 11\%$, $p = 0.043$) for the screws coated with 3 µg testosterone and 2 µg alendronate. In addition, screws coated with a dosage of 9 µg testosterone and 2 µg alendronate resulted in significant increases for BMD ($132 \pm 35\%$, $p = 0.019$), BV/TV ($131 \pm 15\%$, $p = 0.0005$), BS/TV ($123 \pm 11\%$, $p = 0.0009$) and Tb.N ($123 \pm 10\%$, $p = 0.001$). These data demonstrate a dose dependent increase in bone volume, resulting from an increase in trabecular number and not trabecular thickness, reaching significant levels in the highest dose applied (9 µg testosterone). However, this is not reflected in trabecular spacing which is only significantly decreased in the 3 µg testosterone group. This discrepancy may be due to small variations in positioning of the screw in the subchondral bone, as trabecular microstructure will vary within the subchondral compartment.

For VOI2 (Fig 5B) the effects on bone formation were less pronounced but still yielded significant increases in BMD ($122 \pm 24\%$, $p = 0.050$) and BV/TV ($120 \pm 18\%$, $p = 0.012$) for screws coated with the highest amount of testosterone (9 µg). The observation that the combined release of testosterone and alendronate has a significant effect on trabecular bone parameters in such a large area around the screw is unique. Various other types of coatings on implants have been tested in the past, encapsulating for instance hydroxy appetite, BMP2 or BMP2 in combination with TGFβ but generally only showed increases in BV/TV up a maximum of 300 µm from the screw [23]. Most likely this is caused by the fact that large molecules like BMP2 or TGFβ are not able to easily penetrate the surrounding bone tissue. In contrast, it is anticipated that small molecules like testosterone and alendronate easily diffuse into the

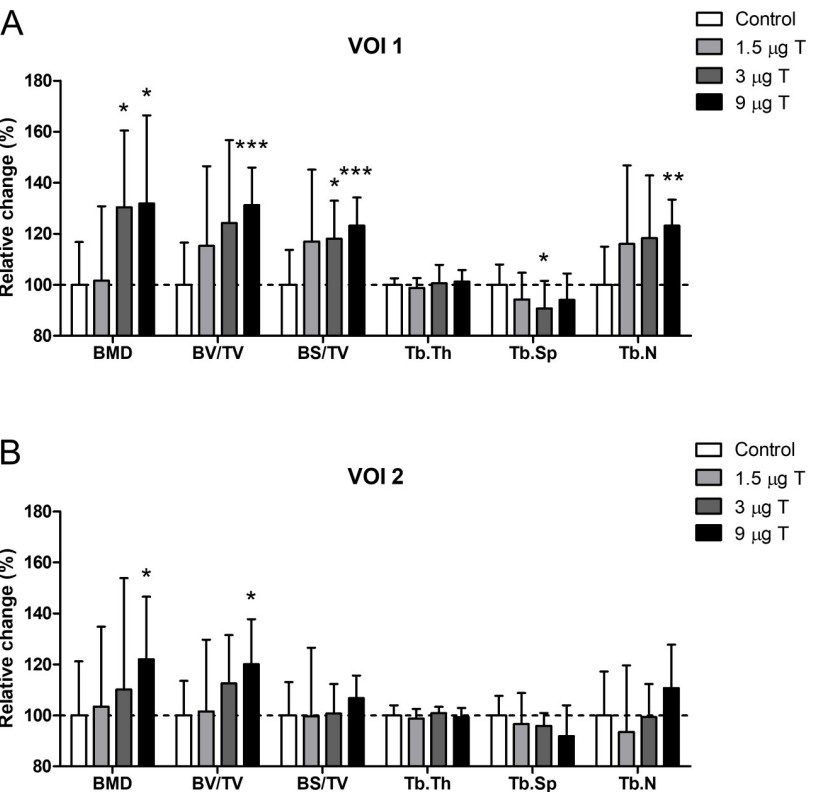

**Fig 5. MicroCT analysis of rabbit femoral condyle implantation model after 8 weeks.** Trabecular bone adjacent to the screw at a distance of 0.3 to 0.8 mm from the screw surface (VOI1), showed dose dependent increases in the following bone parameters, bone mineral density (BMD), bone volume fraction (BV/TV), bone surface fraction (BS/TV) and trabecular number (Tb.N) for testosterone (T) and alendronate (2 µg) coated screws, while trabecular thickness (Tb.Th) and trabecular spacing (Tb.Sp) did not (A, n = 9–10). Trabecular bone adjacent to the screw at a distance of 0.8 to 1.3 mm from the screw surface (VOI2), only showed increased BMD and BV/TV for screws coated with 9 µg testosterone and 2 µg alendronate (B, n = 8–9). Significant differences compared to controls are indicated by asterisks: $^*$p≤0.05, $^{**}$p≤0.01 and $^{***}$p≤0.001.

surrounding bone and, hence, are capable of inducing bone growth at much larger distances from the implant than large protein structures.

Finally, BIC was analyzed on images of the HE stained sections. While no significant increase in BIC was observed for implants coated with 1.5 µg testosterone and 2 µg alendronate, BIC was significantly increased for the implants coated with 3 µg testosterone and 2 µg alendronate (19%, p = 0.014) and 9 µg testosterone and 2 µg alendronate (24%, p = 0.004) as compared to non-coated implants (Fig 6). These data therefore indicate that sustained release of testosterone and alendronate results in enhanced bone regeneration resulting in improved BIC, which is indicative for improved osseo-integration of the implant.

## Conclusions

Controlled or sustained release of drugs at a local site in the human body remains a challenge and is of interest for many therapeutic areas as this potentially overcomes many side effects of drugs normally observed as a consequence of systemic exposure. Here we describe a novel application for local bone regeneration using either pericardial membranes or screws. Both devices were coated with ancillary amounts of 2 bone healing compounds (testosterone and alendronate) embedded in PLGA 5004A as a carrier by ultrasonic spray coating followed by

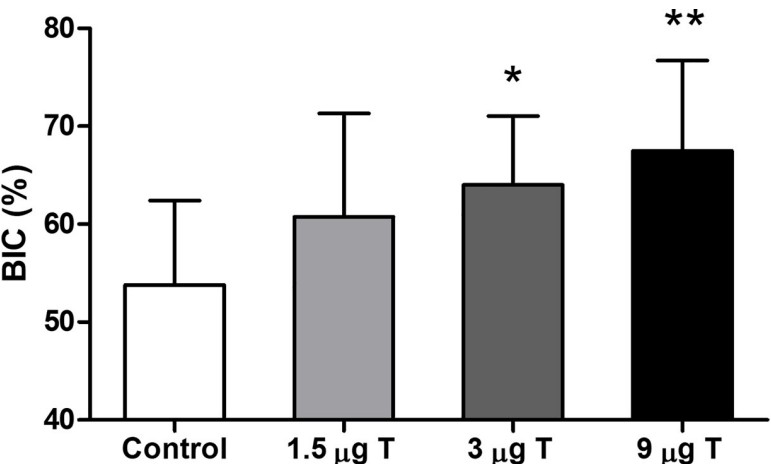

**Fig 6. Bone implant contact (BIC) analysis of rabbit femoral condyle implantation model after 8 weeks.** Coated screws (n = 7–10) showed a dose dependent increase in BIC, with significant increased BIC for screws coated with 3 and 9 μg testosterone (T) and 2 μg alendronate compared to controls. Significant differences compared to controls are indicated by asterisks: *p≤0.05 and **p≤0.01.

final sterilization using gamma irradiation (25 kGy). Application of various top layers of PLGA 5004A, not containing testosterone or alendronate, were essential to prevent a burst release and to achieve in vitro release over a 3-4-week period. In vivo data confirmed that release of these compounds resulted in significant effects on new bone formation as measured by histology and μCT.

In all, our data indicate that local sustained release of testosterone in combination with alendronate may be applied to improve healing of bone defects or fractures or to improve osseo-integration of implants. Importantly, due to the ancillary amounts of testosterone and alendronate required the primary mode of action of a medical device is not affected. Moreover, unlike large biomolecules such as BMP2, testosterone and alendronate are low molecular weight compounds that easily diffuse in bone tissue enabling treatment of large bone segments, although diffusion of alendronate is likely to be lower than diffusion of testosterone due to binding to the bone matrix. In conclusion, the robust bone regeneration properties observed may present a novel and cost-effective approach in the field of bone regeneration.

## Supporting information

**S1 Dataset. Data of figures.** Values, mean and standard deviation of data set used in figures. (XLSX)

## Author Contributions

**Conceptualization:** Jan A. Gossen.

**Investigation:** Cindy J. J. M. van de Ven, Nicole E. C. Bakker, Dennis P. Link, Edwin J. W. Geven.

**Methodology:** Cindy J. J. M. van de Ven, Nicole E. C. Bakker, Dennis P. Link.

**Supervision:** Jan A. Gossen.

**Validation:** Cindy J. J. M. van de Ven, Nicole E. C. Bakker, Dennis P. Link.

**Visualization:** Edwin J. W. Geven.

**Writing – original draft:** Cindy J. J. M. van de Ven.

**Writing – review & editing:** Edwin J. W. Geven, Jan A. Gossen.

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
