## [Decision Letter · Decision Letter 0]

9 Feb 2021

PONE-D-21-00743

Sustained release of ancillary amounts of testosterone and alendronate from PLGA coated pericard membranes and implants to improve bone healing

PLOS ONE

Dear Dr. Gossen,

Thank you for submitting your manuscript to PLOS ONE. After careful consideration, we feel that it has merit but does not fully meet PLOS ONE’s publication criteria as it currently stands. Therefore, we invite you to submit a revised version of the manuscript that addresses the points raised during the review process.

Specifically, both reviewers have raised important concerns regarding inadequacy of data to support the conclusions.  Histology data are not quantitated and analyzed statistically.  The reviewers have also raised a number of concerns that relate to methodology and data presentation. 

We look forward to receiving your revised manuscript.

Kind regards,

Subburaman Mohan

Academic Editor

PLOS ONE

Journal Requirements:

2. Please note that PLOS does not permit references to “data not shown.” Authors should provide the relevant data within the manuscript, the Supporting Information files, or in a public repository. If the data are not a core part of the research study being presented, we ask that authors remove any references to these data.

3.At this time, we request that you  please report additional details in your Methods section regarding animal care, as per our editorial guidelines:

(1) Please state the source of the animals used in the study

(2) Please provide details of animal welfare (e.g., shelter, food, water, environmental enrichment)

(3) Please describe the post-operative care received by the animals, including the frequency of monitoring and the criteria used to assess animal health and well-being.

Thank you for your attention to these requests.

4. Please ensure you have discussed any potential limitations of your study in the Discussion.

5. To comply with PLOS ONE submission guidelines, in your Methods section, please provide additional information regarding your statistical analyses. For more information on PLOS ONE's expectations for statistical reporting, please see https://journals.plos.org/plosone/s/submission-guidelines.#loc-statistical-reporting.

6. Thank you for stating the following in the Financial Disclosure section:

"This work was funded by Osteo-Pharma. JG is CEO of Osteo-Pharma, shareholder and is inventor of patent EP3164133. CvdV and EG are employees of Osteo-Pharma. NB and DL are former employees of Osteo-Pharma."

We note that one or more of the authors are employed by a commercial company: BioConnection

We note that one or more of the authors have an affiliation to the commercial funders of this research study: Osteo-Pharma

c) We note that you have a patent relating to material pertinent to this article. Please provide an amended statement of Competing Interests to declare this patent (with details including name and number), along with any other relevant declarations relating to employment, consultancy, patents, products in development or modified products etc. Please confirm that this does not alter your adherence to all PLOS ONE policies on sharing data and materials, as detailed online in our guide for authors http://journals.plos.org/plosone/s/competing-interests by including the following statement: "This does not alter our adherence to  PLOS ONE policies on sharing data and materials.” If there are restrictions on sharing of data and/or materials, please state these. Please note that we cannot proceed with consideration of your article until this information has been declared.

Reviewers' comments:

Reviewer's Responses to Questions

**Comments to the Author**

1. Is the manuscript technically sound, and do the data support the conclusions?

Reviewer #1: Partly

Reviewer #2: Yes

2. Has the statistical analysis been performed appropriately and rigorously? 

Reviewer #1: Yes

Reviewer #2: Yes

3. Have the authors made all data underlying the findings in their manuscript fully available?

Reviewer #1: Yes

Reviewer #2: Yes

4. Is the manuscript presented in an intelligible fashion and written in standard English?

Reviewer #1: Yes

Reviewer #2: Yes

5. Review Comments to the Author

Reviewer #1: The description of the technology and its application for testosterone are innovative and well-described. However, the conclusions of a possible effects of alendronate using the therapeutic approach are not studied and therefore not supported. Other issues can be with the data presentation and the analysis can be easily addressed. These issues are described in the uploaded review.

Reviewer #2: The manuscript entitled “Sustained release of ancillary amount of testosterone and alendronate from PLGA coated pericard membranes and implants to improve bone healing” described the development of a novel spray coating for slow release of anabolic agent testosterone and bone resorptive inhibitor alendronate from the coated pericardial membrane and screws. Authors optimized the coating and the dosages and tested the release patterns in vitro. They also evaluated the efficacy of testosterone/alendronate coated membrane on bone healing using rat calvaria defects model and bone regeneration of coated screws in rabbit femoral condyle implant models and found the products stimulated calvaria repairing and condyle bone regeneration as evidenced by increased bone volumes, trabecular bone number and reduced trabecular spacing in defected rat calvaria and increased bone formation in implanted rabbit femoral condyle. Overall, the manuscript provided limited in vitro and in vivo data that have not been published previously. Experiments were designed properly, and the statistical analyses were applied. Results supported the conclusion that the testosterone/alendronate coated implants and membranes can be a promising system to induce local bone regeneration and improve healing of bone defects and bone fractures. However, the manuscript could be further improved. Below are some suggestions from this reviewer.

1) Some descriptions in the materials and methods are repeated. For example: 2.3 and 2.4 can be combined. Surgical procedures for rat and rabbit can be combined. Micro-CT and histology parts for rat and rabbit can be combined.

2) Line 151-152. There is an animal welfare issue in this study. Are the rats euthanized with CO2 or overdose of anesthesia, followed by cervical dislocation?

3) Figure 3. It is unclear the increased wound healing resulted from the increased fibrous tissue, or bones or callus. Trichrome staining and safranin O staining are needed to quantify the bone. TRAP staining is also needed to determine if the bone resorption is reduced or not and the data need to be quantified. Is it possible to measure the bone formation markers and bone resorption markers from the healing tissue by either real-time PCR or immunohistochemistry?

4) Figures 5 & 6 can be combined. Fluorescent intensities and areas in Figure 5B in the control and treated sections need to be quantified and statistically analyzed. ntified and statistically analyzed.

6. PLOS authors have the option to publish the peer review history of their article (what does this mean?). If published, this will include your full peer review and any attached files.

Reviewer #1: No

Reviewer #2: **Yes: **weirong xing

---

## [Author Response · Author response to Decision Letter 0]

26 Apr 2021

Editor

We have adjusted the style requirements for the main body and file naming accordingly.

Please note that PLOS does not permit references to “data not shown.”

We have removed the remark “data not shown” from line 216, since the information described here is a general observation and is of no importance for the interpretation of the data presented in the current study.

Please state the source of the animals used in the study. 

We have added the source of that animals (rats and rabbits) to the Materials and Methods section:

“…(Charles Rivers Laboratories)… (Line 121 and line 147)

Please provide details of animal welfare (e.g., shelter, food, water, environmental enrichment)

We have added more details on animal welfare and housing condition of the animals in the Materials and Methods section:

“…and all efforts were made to minimize suffering. Rats were housed under standard laboratory conditions; per pair in standard individual ventilated cages with sawdust bedding, temperature 20-22 °C, 12 h light-dark cycle, relative humidity of 45-55% and ad libitum access to standard rodent chow and water.” (Line 121-124)

“…and all efforts were made to minimize suffering. Rabbits were housed under standard laboratory conditions; housing in groups of 8 with freedom of movement, temperature 20-22 °C, 12 h light-dark cycle, relative humidity of 45-55%, animals were fed twice daily and had ad libitum access to water.” (Line 148-151)

Please describe the post-operative care received by the animals, including the frequency of monitoring and the criteria used to assess animal health and well-being.

We have added more details on post-operative care provided to the animals in the Materials and Methods section.

“Carprofen was given every 24 h for a minimum of 2 days postoperatively. Animals were weighed daily up to 5 days post-surgery and weekly thereafter and were monitored daily for potential signs of dehydration, pain, infection and deviant behavior.” (Line 139-142)

“…and antibiotics were given every 24 h. Animals were weighed daily up to 3 days post-surgery and weekly thereafter and were monitored daily for potential signs of dehydration, pain, infection and deviant behavior.” (Line 163-165) 

Please ensure you have discussed any potential limitations of your study in the Discussion.

An important limitation of this study is the fact that we cannot differentiate between the relative effect of testosterone on osteoblast activation and alendronate on osteoclast inhibition since both molecules are added to the coating. We have therefore added the following text in the Results section:

“Unfortunately, the current experimental design does not allow for the differentiation of the relative effects of osteoblast activation and osteoclast inhibition as no membranes with only testosterone or alendronate were included.” (Line 273-275)

To comply with PLOS ONE submission guidelines, in your Methods section, please provide additional information regarding your statistical analyses. For more information on PLOS ONE's expectations for statistical reporting, please see https://journals.plos.org/plosone/s/submission-guidelines.#loc-statistical-reporting.

We have added the software that was used to statistically analyse the data in this study in the Material and Methods section:

“Statistical analysis was performed using GraphPad Prism 5.02 software.” (Line 203)

 

Reviewer #1

A major issue is the inclusion of alendronate in the study with no characterization, except for one reference to its binding to bone. The half-life of a bisphosphonate on bone is long and this consideration is necessary. 

We agree with the reviewer that the role of alendronate in this study received little attention compared to the role of testosterone. Indeed, a comment that bisphosphonates have a long residency in bone tissue because of binding to hydroxyapatite is relevant. We therefore included the following text in the Introduction: 

“Bisphosphonates bind to hydroxyapatite in the bone matrix and upon bone resorption by osteoclasts inhibits osteoclast activity, binding of bisphosphonates on bone results in prolonged biological half-live, of up to several years [11].” (Line 63-66)

Osteoclast activity would be predicted to be important in the remodeling stages later in bone repair but might actually reduce bone formation early in healing.

We agree with the reviewer that osteoclast activity plays an important role in fracture healing (callus resorption and bone remodelling) and that inhibition of osteoclast activity might therefore negatively affect healing. Several studies have shown an effect on callus remodelling in experimental models upon systemic bisphosphonate treatment, but this did affect fracture healing or mechanical strength. Moreover, clinical studies failed to find a correlation between bisphosphonate treatment and fracture healing in osteoporosis patients. In this sense it is important to note that the amount of alendronate used in this study does completely inhibit osteoclast activity as has been shown in an earlier publication.

To address a possible negative effect of bisphosphonates on fracture healing we added the following text: 

“Beside osteoblast activity, osteoclast activity play an important role in fracture healing, through callus resorption and bone remodelling, and therefore the long-term use of bisphosphonates has been suggested to impair fracture healing. Preclinical studies revealed that systemic bisphosphonate treatment was associated with increased callus size but no delay in fracture healing or decreased mechanical strength was observed [21]. Similarly, in humans, systemic bisphosphonate treatment did not affect fracture healing in distal radius, hip or vertebral fractures [22]. As of such we do not anticipate a negative effect of alendronate on bone formation in this study, considering the local and sustained release of ancillary amounts of alendronate. Indeed, coated membranes with similar amounts of testosterone and alendronate placed over a minipig mandibular defect resulted in increased bone formation, without disrupting bone remodelling as normal osteoblast and osteoclast activity was observed [12].” (Line 287-297) 

Would its diffusion in bone be comparable to testosterone in bone, as suggested at the end of the Discussion?

Although both molecules are small with similar molecular weights, indeed diffusion of testosterone and alendronate may not be similar. Testosterone will diffuse further in the bone tissue compared to alendronate as it will bind to the bone matrix. We therefore added the following text: 

“…, although diffusion of alendronate is likely to be lower than diffusion of testosterone due to binding to the bone matrix.” (Line 368-369) 

For a conclusion of synergistic actions between testosterone and alendronate to be valid, although admittedly only implied by the authors through a reference in the Introduction, their combined effect on healing would have to be greater than the sum of their individual effects. 

We agree with the reviewer that the claim for a synergistic effect of testosterone and alendronate on bone formation is scarcely addressed and should be explained in more detail. The one reference in the introduction that points to a synergistic effect is a study by our group in which we have shown a synergistic effect of the combination of testosterone and alendronate on osteoblast activity (which was greater than the sum of the effect of testosterone or alendronate alone). We therefore added the following text to the Introduction:

 “A synergistic effect of the combination of testosterone and alendronate has been shown in porcine femoral bone biopsies, here the combination of both compounds significantly increased osteoblast activity, which was higher than the sum of stimulation with testosterone and alendronate alone [12].” (Line 66-69)

Additionally, we expanded the text in which other combinations of anabolic and antiresorptive drugs have been shown to have a synergistic effect for fracture healing: 

“Other combination therapies of anabolic and antiresorptive compounds, including BMP2, -7 and parathyroid hormone (PTH) in combination with zoledronate or alendronate, have been shown to synergistically stimulate fracture and calvarial defect healing in preclinical models, as compared to treatment with the anabolic factor or bisphosphonate alone [13-16].” (Line 69-72)

There is no evidence presented in this study that alendronate participates in bone repair (with testosterone) as suggested in the conclusion; it is simply assumed it was released from the membrane or screw implants. As it appears that there are no alendronate (alone) controls or testosterone (alone) controls available, was there a reason that alendronate was included but not characterized in the study?

We agree with the reviewer that a sustained release of alendronate should be addressed in this manuscript, and therefore we have added in vitro release study of alendronate from the membranes that have been used in the rat calvarial defect study and we have included these results in Fig. 2C. Additionally we have added the following text to the Materials and Methods section: 

“Alendronate concentrations from the screw samples were analyzed after a derivatization of alendronate with 2,4-dinitrofluorobenzene (DNFB) at room temperature and spectrophotometric analysis at 374 nm (Lambda 25, PerkinElmer, Waltham, MA, USA) [17].” (Line 108-111) 

And the following text was added to the Results section: 

“Alendronate release was determined in membranes containing 20 µg/cm2 alendronate and a sustained release of alendronate was observed without a burst release, which continued over a period of 28 days (Fig. 2C).” (Line 241-243)

“Alendronate release from screws could not be determined because of the low dose applied to the screws (2 µg) which did not reach the detection limit, however a similar release profile as observed from coated membranes (Fig. 2C) is expected.” (Line 247-249)

The following text was added to the legend of Fig 2:

“.. and alendronate (Aln)..” (Line 444) and “…, while the release of alendronate continues up to 28 days (C, n=4).” (Line 446- 447)

Control groups with only testosterone or alendronate treatment were not included, since the aim of this study was to investigate whether membranes and implants coated with both compounds would result in increased bone formation and not whether this effect is truly synergistic, as its synergistic effect was already established earlier. 

A better description of the data handling is necessary, especially the comparison of the testosterone treatments to the control in the microCT data. There is no mention of the normalization of the various testosterone treatments to the (100%) controls, and the reader is left to conclude it from the figure. This approach is not used for the BIC data analysis.

How were the controls normalized for the comparison with the testosterone treatments by microCT?

Indeed, the normalization of the microCT data is not reported in the Materials and Methods section. We therefore added the following text:

“All values are normalized to the control group, of which the mean is set at 100%.” (Line 187)

Additionally, with respect to this data, the increased trabecular number in the VOI-1 analysis is significant at 9 ug testosterone but there is no difference in the trabecular spacing. Likewise, the trabecular spacing in the 3 ug testosterone VOI-1 analysis is significantly reduced but there is no difference in the trabecular thickness or trabecular number parameters at this dosage. It would be expected that an increase in trabeculae would decrease their spacing for 9 ug testosterone, and a decrease in their spacing for 3 ug testosterone would be due to an increase in their number or thickness, or both. Is there an explanation for this observation, possibly related to the model or the analysis approach?

We agree with the reviewer that the results of the microCT analysis of VOI1 are not completely consistent. The dose dependent increase in bone volume is clearly due to a dose dependent increase in trabecular number, and not trabecular thickness, although it only reaches a significant difference at the highest dose. One would indeed expect a dose dependent decrease of trabecular spacing, however this is not clearly observed, although all experimental groups show a decreased trabecular spacing, and only trabecular spacing of the 3 µg T group is significantly reduced. 

First, it is important to note that reductions are small and the significancy level of the 3 ug T group is just under 0.05 (p=0.043). A reason for this discrepancy is likely due to small differences in the positioning of the screw in the subchondral bone. The trabecular structure in the subchondral compartment is not homogeneous and trabecular spacing will differ at varying positions in the subchondral bone, which may be responsible for the lack of significant reduction of trabecular spacing in the 9 ug T group. To address and explain this discrepancy we added the following text:

“These data demonstrate a dose dependent increase in bone volume, resulting from an increase in trabecular number and not trabecular thickness, reaching significant levels in the highest dose applied (9 µg testosterone). However, this is not reflected in trabecular spacing which is only significantly decreased in the 3 µg testosterone group. This discrepancy may be due to small variations in positioning of the screw in the subchondral bone, as trabecular microstructure will vary within the subchondral compartment.” (Line 327-333)

Insertion of the figure legends within the text of the manuscript interrupts the description it and makes it more difficult to follow.

The figure legends are now placed separately at the end of the manuscript.

The description of the anesthesia in lines 183 through 187 is unclear. Two anesthesia approaches are described. Was isoflurane inhalation used in addition to injection anesthesia?

Indeed, the anesthesia protocol is not described accurately, propofol and atropine were not used in this study, instead ketamine/dexmedetomidine was used to induce anesthesia and inhalation of isoflurane to maintain anesthesia. The following text in the Materials and Methods section was adjusted accordingly:

“Anesthesia was induced by an intramuscular injection of ketamine (10 mg/kg, Alfasan) and dexmedetomidine (100 µg/kg, Orion Pharma) and was maintained by inhalation of isoflurane combined with oxygen. After surgery, anesthesia was antagonized by antisedan (5 mg/ml, Orion Pharma).” (Line 151-154)

Providing the dynamic bone histomorphometry in Figure 5 prior to the microCT data in Figure 4 would introduce the two-VOI approach before the figure in which it is used. The microCT analysis always precedes the histology with respect to the samples but inverting the presentation of these figures in the manuscript would clarify the two-VOI approach for the microCT analysis.

We agree with the reviewer that by changing the order of Fig 4 and Fig 5, this will enhance the quality of the dataset, as this will clarify the VOI approach applied in the µCT analysis. We changed the order of the figures and the description in the results accordingly.

Reviewer #2

Some descriptions in the materials and methods are repeated. For example: 2.3 and 2.4 can be combined. Surgical procedures for rat and rabbit can be combined. Micro-CT and histology parts for rat and rabbit can be combined.

We agree that several parts of the Materials and Methods section can be combined, resulting in a more concise description. Parts in which information was combined are: “In vitro release of testosterone and alendronate”, “Surgical procedures”, “MicroCT analysis” and “Histology”.

Line 151-152. There is an animal welfare issue in this study. Are the rats euthanized with CO2 or overdose of anesthesia, followed by cervical dislocation?

Indeed, the rats were not just euthanized by just cervical dislocation. First the rats were euthanized by CO2 followed by cervical dislocation. We therefore added the following text: 

“…by CO2 followed…” (Line 144)

Figure 3. It is unclear the increased wound healing resulted from the increased fibrous tissue, or bones or callus. Trichrome staining and safranin O staining are needed to quantify the bone. TRAP staining is also needed to determine if the bone resorption is reduced or not and the data need to be quantified. Is it possible to measure the bone formation markers and bone resorption markers from the healing tissue by either real-time PCR or immunohistochemistry?

We understand the reviewer’s concern on the difference in defect healing between the control and membrane covered defects. Regarding the different tissues involved in wound healing, the main focus of this study is the effect of the coating on bone formation and not on the formation of fibrous tissue and is hence not considered of importance within the scope of this study. Since no bone is present in the defect at t=0 all signal quantified by microCT is mineralized new bone formed during healing. Indeed, endochondral ossification will precede mineralization of the new bone and Trichrome or SafO staining will give more insight in the ratio of cartilaginous and mineralized bone. But again, we believe that such histological quantification is not required regarding the aim of the study, as functional (mineralized) bone is our primary outcome parameter. We therefore have not included Trichrome of SafO staining but we have addressed the fact that only mineralized bone can be quantified by microCT by adding the following text:

“…mineralized..” (Line 268 and 271)

With regard to the request for quantification of bone resorption by TRAP staining, we do not think that such staining will provide more insight if bone resorption in decreased. Firstly, since alendronate concentration on both membranes is similar and secondly because bone formation follows a very different pattern between control and coated membrane covered defects. In control membranes bone is formed at the edge of the defect while in membrane covered defect new bone formation is associated with the membrane, as a result osteoclast activity will inherently differ between groups, making it impossible to interpret these results as a reduction in bone resorption. The same applies for bone formation markers. We therefore have not performed a TRAP staining or measured other bone resorption or bone formation markers by histology or PCR, but we did address the difference in bone formation between the groups in more detail. Moreover, we addressed the limitation of the study design since no membranes were included with only testosterone or alendronate and added the following text in the Result section: 

“Clearly, bone formation in the control and membrane covered defects follows different bone remodelling patterns which is likely reflected by different dynamics in osteoblast and osteoclast activity. Unfortunately, the current experimental design does not allow for the differentiation of the relative effects of osteoblast activation and osteoclast inhibition as no membranes with only testosterone or alendronate were included.” (Line 271-275) 

Figures 5 & 6 can be combined. Fluorescent intensities and areas in Figure 5B in the control and treated sections need to be quantified and statistically analyzed. 

We understand the suggestion to combine Fig 5 and 6, however since reviewer 1 has suggested to change the order of Fig 4 and 5, as this would clarify the two-VOI approach for the µCT analysis, we do not want to combine Fig. 5 and 6, as we believe Fig. 6 (BIC) is the most clinically relevant data set on osteointegration and should therefore be presented at the end of the manuscript. 

We agree with the reviewer that the observation of just more fluorescent signal in a single histological section is subjective and insufficient to make this claim we indeed need to quantify the fluorescent signal in the described. We therefore performed a quantification of the % of signal of calcein green and alizarin complexone in the described regions in the control group and in the group with coated screws. However, no significant difference in % of signal of calcein green and alizarin complexone in ROI1 and ROI2 was observed, and we therefore have to reject the claim that the coated screw results in increased incorporation of fluorescent dyes at week 2 or week 4 after implantation. We have not included these data in an extra figure, but we have described the results in detail in the text. And we have removed a histological picture from Fig 4, now only showing 1 illustrative picture, since showing a fluorescent section of both groups would suggest a difference in signal. We have adjusted the following texts.

In the Material and Methods section:

“Unstained sections were analyzed by fluorescence microscopy and fluorescent signal of calcein green and alizarin complexone was determined in a region of interest (ROI) positioned at 0.3-0.8 mm and 0.8-1.3 mm from the screw surface using ImageJ software.” (Line 199-201)

In the Results section:

“The fluorescent signal of both dyes was analyzed on sections of non-coated implants and implants coated with 9 µg testosterone and 2 µg alendronate. Signal was determined in ROIs positioned at 0.3-0.8 mm (ROI1) and 0.8-1.3 mm (ROI2) from the screw surface and percentage of fluorescent signal to total ROI area was calculated. No significant difference in fluorescent signal of calcein green in ROI1 (control; 4.1 + 3.1%, 9 µg T; 4.3 + 2.1%, p=0.908) or ROI2 (control; 4.5 + 2.8%, 9 µg T; 2.4 + 2.2%, p=0.356) and of alizarin complexone in VOI1 (control; 5.9 + 8.9%, 9 µg T; 2.2 + 1.5%, p=0.517) or VOI2 (control; 3.9 + 5.6%, 9 µg T; 0.8 + 0.4%, p=0.0.932) was observed. The lack of a significant difference in incorporation fluorescent dyes at week 2 and week 4 after implantation may be explained by large standard deviation in fluorescent signal which may be due to small variations in the position of the screw in the subchondral bone and the position of the histological section that is analyzed. Moreover, the fluorescent signal only reflects bone formation at a single time point and may not be representative for the total bone formed at end point.” (Line 307-319)

In the Legends of Fig. 4:

“Incorporation of calcein green (green signal) and alizarine complexone (red signal) fluorescent dyes into newly formed bone at 2 and 4 weeks after implantation, respectively, of a screw coated with 9 µg testosterone and 2 µg alendronate (B). Fluorescent signal was observed directly at the surface of the screw (dashed line) but also extending into VOI1 (0.3-0.8 mm from screw) and VOI2 (0.8-1.3 mm from screw).” (Line 464-468)

---

## [Editor Report · Decision Letter 1]

5 May 2021

Sustained release of ancillary amounts of testosterone and alendronate from PLGA coated pericard membranes and implants to improve bone healing

PONE-D-21-00743R1

Dear Dr. Gossen,

We’re pleased to inform you that your manuscript has been judged scientifically suitable for publication and will be formally accepted for publication once it meets all outstanding technical requirements.

Kind regards,

Subburaman Mohan

Academic Editor

PLOS ONE
---

## [Editor Report · Acceptance letter]

7 May 2021

PONE-D-21-00743R1 

Sustained release of ancillary amounts of testosterone and alendronate from PLGA coated pericard membranes and implants to improve bone healing 

Dear Dr. Gossen:

I'm pleased to inform you that your manuscript has been deemed suitable for publication in PLOS ONE. Congratulations! Your manuscript is now with our production department. 

Kind regards, 

on behalf of

Dr. Subburaman Mohan 

Academic Editor

PLOS ONE